# Fe Speciation in Iron Modified Natural Zeolites as Sustainable Environmental Catalysts

**Fernando Chávez Rivas [1], Inocente Rodríguez-Iznaga [2,*], Gloria Berlier [3,*], Daria Tito Ferro [4], Beatriz Concepción-Rosabal [2] and Vitalii Petranovskii [5]**

[1] Instituto Politécnico Nacional, ESFM, Departamento de Física, UPALM, Zacatenco, Ciudad de México 07738, Mexico; fchavez@esfm.ipn.mx

[2] Instituto de Ciencia y Tecnología de Materiales (IMRE)-Universidad de La Habana, Zapata y G s/n, La Habana, C.P. 10400, Cuba; beatriz@imre.uh.cu

[3] Dipartimento di Chimica and NIS Centre, Università di Torino, Via P. Giuria 7, 10125 Torino, Italy

[4] Centro Nacional de Electromagnetismo Aplicado (CNEA), Universidad de Oriente, Ave. Las Américas s/n, Santiago de Cuba, C.P. 90400, Cuba; dariat@uo.edu.cu

[5] Departamento de Nanocatálisis, Centro de Nanociencias y Nanotecnología (CNyN), Universidad Nacional Autónoma de México, Carretera Tijuana-Ensenada, Km 107. Ensenada, C.P. 22860 BC, Mexico; vitalii@cnyn.unam.mx

\* Correspondence: inocente@imre.uh.cu (I.R.-I.); gloria.berlier@unito.it (G.B.)

**Abstract:** Natural purified mordenite from Palmarito de Cauto (ZP) deposit, Cuba, was subjected to a hydrothermal ion exchange process in acid medium with $Fe^{2+}$ or $Fe^{3+}$ salts ($Fe^{2+}$ZP and $Fe^{3+}$ZP). The set of samples was characterized regarding their textural properties, morphology, and crystallinity, and tested in the NO reduction with $CO/C_3H_6$. Infrared spectroscopy coupled with NO as a probe molecule was used to give a qualitative description of the Fe species' nature and distribution. The exchange process caused an increase in the iron loading of the samples and a redistribution, resulting in more dispersed $Fe^{2+}$ and $Fe^{3+}$ species. When contacted with the NO probe, $Fe^{2+}$ZP showed the highest intensity of nitrosyl bands, assigned to NO adducts on isolated/highly dispersed $Fe^{2+}/Fe^{3+}$ extra-framework sites and $Fe_xO_y$ clusters. This sample is also characterized by the highest NO sorption capacity and activity in NO reduction. $Fe^{3+}$ZP showed a higher intensity of nitrosonium ($NO^+$) species, without a correlation to NO storage and conversion, pointing to the reactivity of small $Fe_xO_y$ aggregates in providing oxygen atoms for the NO to $NO^+$ reaction. The same sites are proposed to be responsible for the higher production of $CO_2$ observed on this sample, and thus to be detrimental to the activity in NO SCR.

**Keywords:** natural zeolite; mordenite; Iron exchange; FTIR-NO; HRTEM; NO reduction

## 1. Introduction

Transition metal ion-exchanged zeolites are being considered for practical applications in diesel and stationary combustion sources' emission control due to good stability at high temperatures and other qualities. Among other metals, iron has shown interesting catalytic properties and it is particularly attractive due to its low cost and non-toxic nature [1]. The activity of Fe-zeolites in the selective catalytic reduction (SCR) of $NO_x$ has been studied in two main processes, i.e., with ammonia ($NH_3$-SCR) [2–8] and hydrocarbons (HC-SCR) as reducing agents. In the latter case, a variety of compounds, such as $C_3H_6$ [9], iso and n-$C_8H_{18}$ [10], $C_3H_8$ [11], iso-$C_4H_{10}$ [12], and $C_2H_5OH$ [13], have been evaluated. Even though $NH_3$-SCR is recognized among the most efficient $deNO_x$ processes, HC-SRC has the main advantage of using a gas mixture very similar to that found in exhausts [14,15], avoiding the use and related safety/environmental concerns of $NH_3$ sources. Natural zeolites, such as mordenite, have

also drawn attention as sustainable materials due to their ion exchange properties, thermal stability, availability, and low price. Therefore, efficient and low-cost catalysts could be developed from natural mordenite modified with transition metal ions, such as iron.

Several types of iron species, such as isolated or binuclear ions, and iron oxide clusters stabilized in zeolite pores, have been proposed as active sites in $NO_x$ and $N_2O$ decomposition processes. Rutkowska et al. proposed that monomeric $Fe^{3+}$ cations and small $Fe_xO_y$ oligomers in Fe-beta zeolites are more active than larger iron oxide aggregates in the catalytic decomposition and reduction of $N_2O$ [16]. Kim et al. pointed to the activity of small $Fe^{2+}$-enriched iron oxide clusters in zeolite pores, maximized by reduction with hydrogen, in the same reaction [17]. Similar oligonuclear $Fe_xO_y$ clusters, coupled to isolated $Fe^{3+}$ ions, were reported to be active in Fe-containing AlPO-5, where $N_2O$ SCR was carried out with $CH_4$ in the presence of water vapor and oxygen [18]. The authors also pointed out that larger $Fe_2O_3$ particles are able to activate both SCR and methane combustion only at temperatures higher than 500 °C. Fe-exchanged MOR, FER, BETA, and ZSM-5 prepared with iron (III) acetylacetonate precursor were shown to present a mixture of $Fe^{3+}$ and $Fe^{2+}$, with catalytic activity in the NO reduction with $C_3H_6$ dependent on the $Fe^{2+}/Fe^{3+}$ ratio, especially at low temperatures [19]. Fe/MOR showed the highest $NO_x$ adsorption capacity and catalytic activity in the 200 to 400 °C temperature range. These few (not exhaustive) examples give a taste about the complexity of iron species that can be formed within zeolite pores. Zeolite's inner (and external) surface is indeed intrinsically rich in sites (negatively charged framework oxygen atoms, defective silanols, and silanol nests) able to stabilize Fe sites with a variety of coordination, nucleation, and oxidation states, strongly affecting their catalytic activity and selectivity [20].

Our team has earlier reported the characterization of a purified natural mordenite from Palmarito de Cauto deposit (Cuba), ion-exchanged by hydrothermal treatments with $Fe^{2+}$ and $Fe^{3+}$ salts. The three samples (parent ZP, $Fe^{2+}$ZP, and $Fe^{3+}$ZP, see the experimental section for details) were characterized by diffuse reflectance UV-Vis and Mössbauer spectroscopies [21]. These techniques showed the presence of a large variety of iron species already in the not-exchanged parent material: Highly dispersed Fe ions in extra framework positions, octahedral Fe ions in oligomeric clusters of the $Fe_xO_y$ type inside the channels, and larger iron oxide/hydroxide clusters and $Fe_2O_3$ particles located on the external surface of the zeolite crystals. After the hydrothermal exchange, the presence of $Fe^{2+}$ and $Fe^{3+}$ in the $Fe^{2+}$ZP and $Fe^{3+}$ZP samples, respectively, was confirmed by Mössbauer and UV-Vis. Moreover, iron oxide agglomerates are favored in $Fe^{3+}$ZP, and dispersed cationic species in $Fe^{2+}$ZP. Together with a plethora of oxy-hydroxides clusters/aggregates, $Fe^{2+}$ were proposed to be present in extra-framework positions as a charge compensating cations. The formation of Brønsted sites, likely formed by insertion of $Fe^{3+}$ ions in framework positions during hydrothermal treatment, was moreover assessed by infrared spectroscopy in $Fe^{3+}$ZP.

The use of natural zeolites as catalyst has an intrinsic limitation related to phase purity, which cannot be controlled as carefully as for synthetic ones [22–25]. This also implies that the interpretation of the results can be very complex. On the other hand, these are relatively low-cost catalyst materials and importantly, no templates need to be used for their preparation, thus lowering the environmental impact of the process. With this respect, we here report about a detailed physico-chemical characterization of the set of samples ZP, $Fe^{2+}$ZP, and $Fe^{3+}$ZP, trying to correlate the nature and distribution of present Fe species to the NO storage and conversion in the $CO/C_3H_6$ SCR of NO, which is used as a test reaction.

## 2. Results

### 2.1. Elemental Analysis

The iron content of the starting material ZP, and its exchanged forms $Fe^{2+}$ZP and $Fe^{3+}$ZP, are 1.55%, 3.06%, and 2.68%, respectively. As discussed with more detail in our earlier paper [21], iron is already present in the natural zeolite sample. However, after the hydrothermal exchange treatments,

the amount of iron increases, as a result of ion exchange of $Fe^{2+}$ and $Fe^{3+}$ with cations present in the natural mordenite, mainly $Ca^{2+}$ and $Na^+$.

## 2.2. Catalytic Activity in HC-SCR

The catalytic activity of the set of samples in the NO reduction with $CO/C_3H_6$ in the presence of oxygen was measured as a function of temperature, as summarized in Figures 1 and 2. Figure 1 shows the change in the amount of NO and $CO_2$ in the gas output, monitored from 25 up to 500 °C. Figure 2 reports the NO conversion, which is negligible below 170 °C for all samples. Before carrying out the catalytic test, a steady state concentration of the gas in the flow was reached, around 700 ppm for NO at the reactor outlet. Once this condition was reached, the increase of temperature caused the desorption of the NO absorbed on the catalyst's surface. This can be seen in Figure 1a as a positive peak between RT and 200 to 250 °C. In this interval, no NO conversion is thus observed in Figure 2. Above this temperature, the negative peaks in Figure 1a correspond to NO conversion, as described below. It is well known that adsorption is the primary step in a catalytic reaction. Accordingly, studies reported on the catalytic test with Fe-MOR [19] and Cu-MFI [26] showed that these catalysts both have great capacities for NO adsorption and high catalytic activity for the selective reduction of NO. In agreement with these reports, the amount of NO absorbed/desorbed during the first step of NO interaction (from RT to 200 °C, Figure 1a) is related to the adsorption capacity of the materials. Thus, $Fe^{2+}ZP$ shows a higher NO adsorption capacity with respect to $Fe^{3+}ZP$ and, even more, with the ZP sample (full, dashed and dotted lines, respectively).

Coming to the different catalytic performances of the samples, ZP shows the lowest activity, with a low conversion (8%) starting from 170 °C, as shown in Figure 2. On the contrary, the highest activity is observed for $Fe^{2+}ZP$ sample (34%), followed by $Fe^{3+}ZP$ (19%), indicating the importance of the ion exchange hydrothermal process in order to increase the population of active sites. The conversion of $Fe^{2+}ZP$ is comparable to literature data obtained in similar conditions [19]. Moreover, $Fe^{2+}ZP$ shows a reaction onset below 190 °C while $Fe^{3+}ZP$ only starts to be significantly active in NO reduction at around 240 °C.

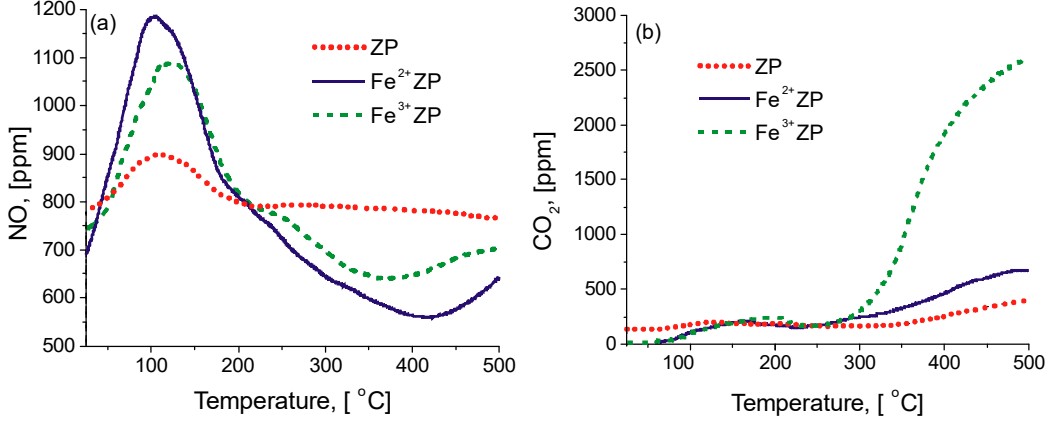

**Figure 1.** Behavior of the catalytic desorption–reduction of NO with $CO/C_3H_6$ and $O_2$ (**a**) and corresponding $CO_2$ formation (**b**) over the parent zeolite (ZP) and exchanged samples as function of temperature.

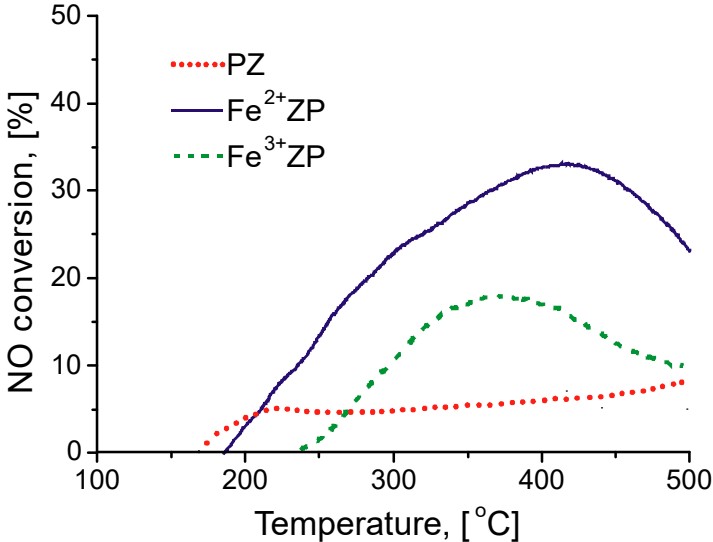

**Figure 2.** Dependence of NO conversion (%) with $CO/C_3H_6$ on the parent zeolite (ZP) and the exchanged samples as function of reaction temperature.

The formation of $CO_2$ during NO reduction with $CO/C_3H_6$ was also followed, and is presented in Figure 1b. The highest $CO_2$ production is observed on sample $Fe^{3+}ZP$, while $Fe^{2+}ZP$ and ZP have similar $CO_2$ productivity. This trend is not correlated to the zeolite's activity in NO reduction, and can be explained by a direct oxidation of CO and $C_3H_6$ by $O_2$. This suggests that sample $Fe^{3+}ZP$, only mildly active in NO reduction, is instead a good oxidation catalyst. This reactivity could be related to the presence of small $Fe_xO_y$ aggregates, providing oxygen atoms and reversible electron transfer involving $Fe^{3+}/Fe^{2+}$ ions. The formation of these species after $Fe^{3+}$ insertion in parent ZP was observed by UV-Vis spectroscopy in our previous report [21]. Thus, CO and/or $C_3H_6$ oxidation may be one of the reasons for the lower activity in NO reduction of $Fe^{3+}ZP$ with respect to $Fe^{2+}ZP$.

Although the NO conversion levels achieved by using natural zeolite-based catalysts (included those studied in this work) are not elevated, the related studies provide important knowledge on the iron speciation in the samples and on their reactivity, as discussed in the following where a detailed physico-chemical characterization of the materials is reported.

### 2.3. XRD Analysis

The XRD patterns of the samples are shown in Figure 3. In parent ZP, the diffraction peaks of the mordenite structure are present, as can be seen in the figure where they are indexed to the (hkl) corresponding crystallographic planes. The diffraction pattern also shows peak related to impurity phases (quartz and clinoptinolite), as labeled in Figure 3. A small change in the intensity of the peaks is observed on $Fe^{2+}ZP$ and $Fe^{3+}ZP$, as is often observed after ion exchange as a consequence of changes in the location and concentration of extra-framework cations in the zeolite channels [27,28]. This suggests that the crystalline structure of mordenite was not altered by the treatment with acid solutions, and that the zeolite is stable under applied conditions.

Besides the changes in intensity mentioned above, a small negative shift (about 0.14 degrees in average) with respect to parent ZP is observed for the position of most of the peaks of sample $Fe^{3+}ZP$, which is instead negligible for $Fe^{2+}ZP$ (Table 1). This corresponds to an increase of the distances between crystallographic planes, and can be associated to an expansion of the zeolite matrix due to the incorporation of iron into the mordenite framework [29,30]. Indeed, this observation is in very good agreement with infrared evidences obtained for the formation of Brønsted Fe(OH)Si sites after hydrothermal treatment with $Fe^{3+}$ [21].

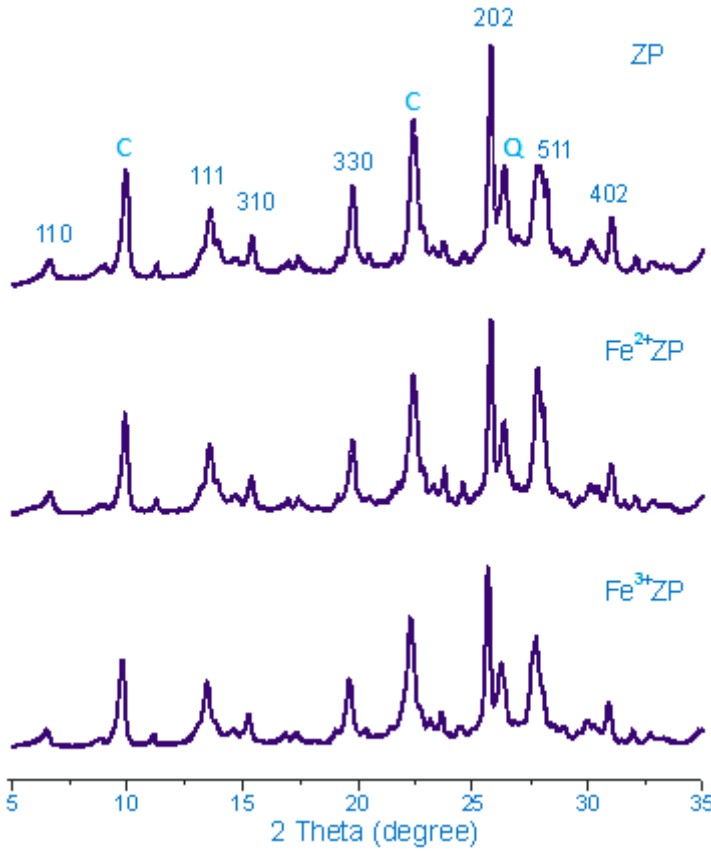

**Figure 3.** XRD patterns of the samples, with hkl labels of the mordenite structure. Q: quartz, C: clinoptilolite.

**Table 1.** Peaks position analysis of sample ZP and its exchange form, with the measured negative shift in brackets.

| Plane | Peak Position (Shift) | | |
|:---:|:---:|:---:|:---:|
| | ZP | $Fe^{2+}ZP$ | $Fe^{3+}ZP$ |
| 111 | 13.57 | 13.55 (0.02) | 13.44 (0.11) |
| 310 | 15.41 | 15.40 (0.01) | 15.27 (0.13) |
| 330 | 19.77 | 19.74 (0.03) | 19.59 (0.15) |
| 202 | 25.76 | 25.76 | 25.63 (0.13) |
| 402 | 31.06 | 31.04 (0.02) | 30.89 (0.15) |

*2.4. Textural Properties*

The textural properties of the three samples, determined by low-temperature nitrogen sorption measurements, are summarized in Table 2. A small increase in the specific surface area is observed passing from parent ZP to its exchanged forms. This trend can be related to the used exchange process, likely removing impurities or extra-phases (e.g., amorphous volcanic glass) from the zeolite particles' external surface. Moreover, the small increase in total surface area is accompanied by a mild decrease in the micropores' surface area and volume (see last columns of Table 2), in agreement with the formation of low solubility iron oxy-hydroxides phases inside the mordenite channels, as testified by our previous work [21]. This could be explained by a reaction of exchanged iron cations with atmospheric oxygen and OH- groups retained (adsorbed or occluded) in the zeolite pores, which are liberated during the ion exchange. Numerous studies are available in the literature on the occurrence of these processes, mainly during ion exchange [31–35]. Also, $Fe^{3+}$ hydrolysis is expected to occur to a greater extent than that of $Fe^{2+}$ [36,37]. However, we acknowledge that the observed changes are very small, being very

close to the experimental error of 5%, so that formation of occluded particles should be limited to a restricted amount of small clusters/aggregates, in agreement with the absence of extra peaks in the XRD patterns of exchanged samples. No evidence for the formation of mesoporosity (often observed after acid treatment of zeolites) was observed, in agreement with the high stability of mordenite in these conditions.

**Table 2.** Textural properties of the studied samples.

| Sample | $SSA_{BET}$ (m$^2$/g) | $SSA_{Lang}$ (m$^2$/g) | $SSA_{ext}$ (m$^2$/g) | $V_{mic}$ (cm$^3$/g) | $A_{mic}$ (m$^2$/g) |
|---|---|---|---|---|---|
| ZP | 244 | 312 | 3 | 0.109 | 241 |
| $Fe^{2+}$ZP | 259 | 331 | 26 | 0.105 | 233 |
| $Fe^{3+}$ZP | 256 | 328 | 26 | 0.104 | 230 |

$SSA_{BET}$ = Specific surface area calculated using the standard Brunauer–Emmet–Teller (BET) method. $SSA_{Lang}$ = Specific surface area, calculated from Langmuir method. $SSA_{ext}$ = External surface area, calculated using the t-plot method. $V_{mic}$ = Micropore volume, calculated using the t-plot method. $A_{mic}$ = Micropore surface area, calculated using the t-plot method.

*2.5. HRTEM Analysis*

High-resolution TEM was used to investigate the effect of hydrothermal ion exchange on the morphology of the parent natural purified mordenite ZP. Measurements were carried out at 100 kV in order to avoid beam-induced damaging of the zeolite particles [38]. Figure 4 shows the TEM images of ZP sample, which is composed of particles of irregular shape, with sizes in the range of 200 to 400 nm, formed by the agglomeration of smaller particles. The left-hand panel of Figure 4 shows a magnification, allowing us to appreciate the crystalline nature of the zeolite. The measured lattice distance is 0.38 nm, in agreement with the (241) plane of mordenite [39].

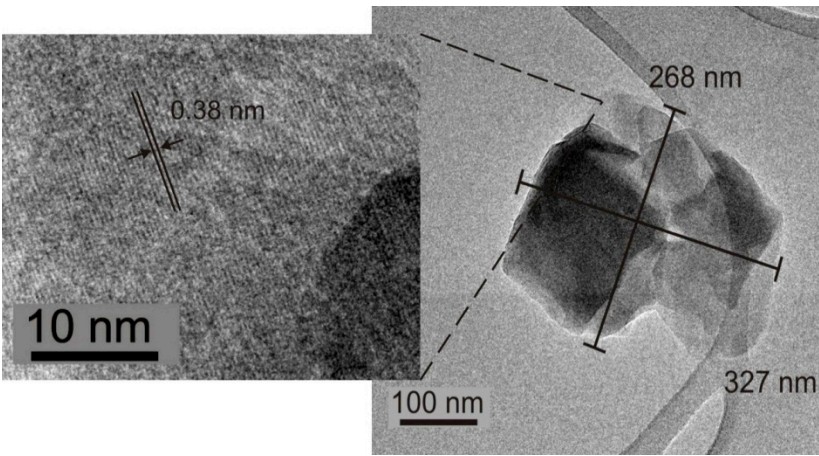

**Figure 4.** HRTEM images of ZP, with magnification of a particle border allowing appreciation of the mordenite lattice planes.

Representative TEM images of $Fe^{2+}$ZP and $Fe^{3+}$ZP samples are shown in Figures 5 and 6, respectively. In both cases, the morphology of the main particles is similar to that of ZP sample, with relatively large agglomerates of regular crystalline particles. Smaller particles (1 to 5 nm) can be appreciated in the figure magnifications, which could be related to the clinoptilolite extra phase on the basis of the measured crystallographic fringes.

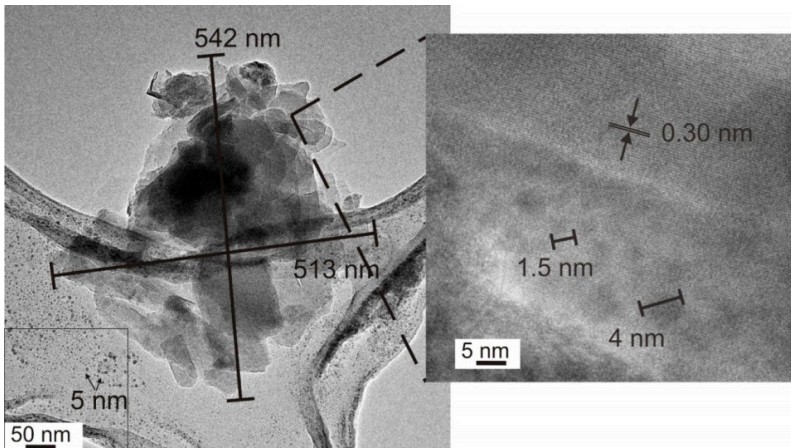

**Figure 5.** HRTEM images of $Fe^{2+}$ZP sample. Arrows in the bottom part of left-hand panel indicate the presence of small particles of extra phases. Some of these can be also seen in the right-hand magnification, where particles' size and crystalline fringes' distances were measured.

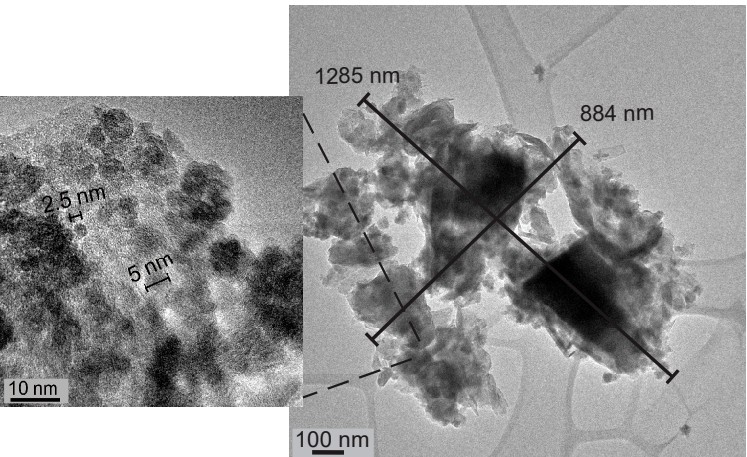

**Figure 6.** HRTEM images of $Fe^{3+}$ZP sample. The left-hand panel reports magnifications of the central image, showing the presence of nanometric-sized particles, likely related to the clinoptilolite extra phase.

EDS microanalysis carried out at different magnifications indicates a large heterogeneity in the iron distribution in all samples. Moreover, this analysis (spectra not reported) shows a difference in the cation exchange capability: While ZP and $Fe^{3+}$ZP samples show the presence of Mg, Na, K, Ca, and Fe, the $Fe^{2+}$ZP sample only presents Ca and Fe, in agreement with previous reports [21].

*2.6. NO Adsorption Studies Followed by FTIR*

The adsorption of NO or $NO_2$ on acid- and metal-exchanged zeolites followed by infrared spectroscopy is a well-known method for the characterization of exposed ions, through the formation of adsorbed complexes with characteristic vibrational fingerprints. Moreover, indirect indications about the Brønsted or metal ions' reactivity can be inferred by observing the formation of adsorbed NO/$NO_2$ reaction products, such as nitrosonium ions $NO^+$, nitrates $NO_3^-$, nitrites $NO_2^-$, and/or water [40–44].

NO adsorption experiments were carried out at room temperature (RT) on ZP, $Fe^{2+}$ZP, and $Fe^{3+}$ZP, after oxidation and reduction treatments. Modifications in the spectra upon dosing NO were observed in the OH stretching ($\upsilon$OH, from 3800 to 3000 $cm^{-1}$), in the nitrosonium and nitrosyl (from 2300 to 1700 $cm^{-1}$), and in the water bending $\delta(H_2O)$ and nitrate (from 1750 to 1300 $cm^{-1}$) regions. Spectra discussion will be mainly focused on the nitrosonium/nitrosyl region, showing the most intense and informative bands.

### 2.6.1. ZP Sample

The results obtained on the oxidized and reduced ZP sample are shown in Figure 7A and B, respectively. On oxidized ZP, three groups of bands are present: One centered at 2265 cm$^{-1}$, a second between 2200 and 1920 cm$^{-1}$, and a third one between 1920 and 1700 cm$^{-1}$. Some of these are also present in the reduced sample (bottom), though with lower intensity. The first group is formed by an intense band at 2265 cm$^{-1}$, with shoulders at 2282 and 2251 cm$^{-1}$, which can be assigned as N$_2$O formed upon reaction with NO on different Fe$^{x+}$ ($x$ = 2, 3) surface sites on small iron oxide particles [43,45–47].

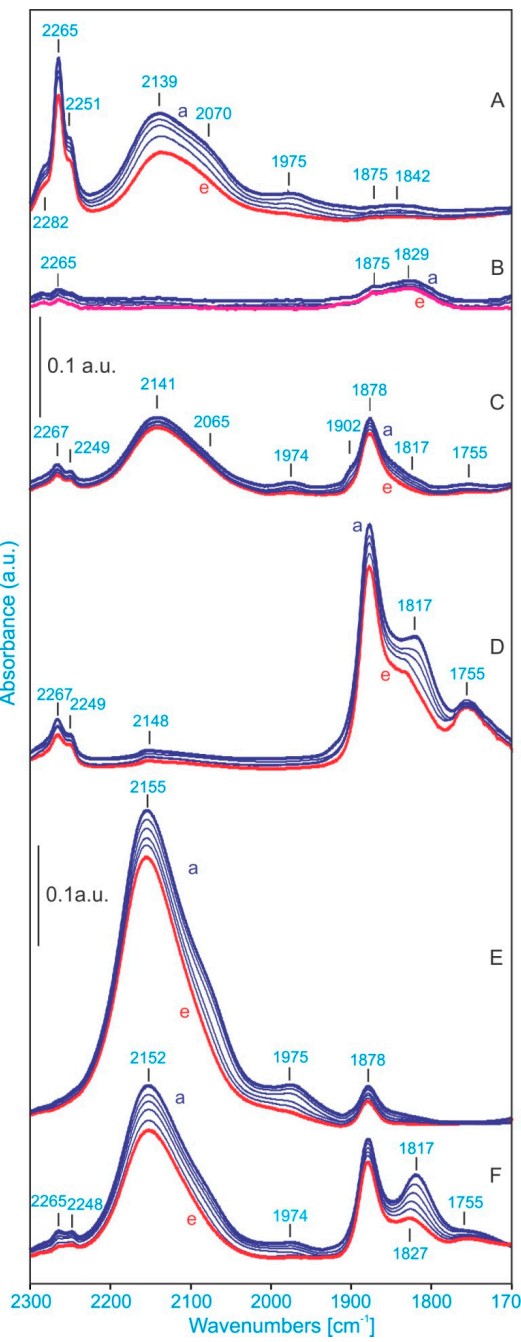

**Figure 7.** Infrared spectra in the nitrosonium/ nytrosyl region of NO adsorbed at RT on oxidized (**A**) and reduced (**B**) ZP sample, oxidized (**C**) and reduced (**D**) Fe$^{2+}$ZP sample, and oxidized (**E**) and reduced (**F**) Fe$^{3+}$ZP sample. Equilibrium P$_{NO}$ = 0.4 mBar (a) and subsequent stepwise evacuation up to dynamic vacuum (e).

Coming to the second group of bands, it is composed of a broad absorption, with a maximum at 2139 cm$^{-1}$ and a shoulder at 2070 cm$^{-1}$, with a very broad band centered at 1975 cm$^{-1}$, which are not present on the reduced sample. Similar bands have been assigned by different authors to NO$^+$ [48–50], formed by the reaction of NO with adsorbed oxygen atoms, and with the formation of NO$_2$, which then reacts with protons to give NO$^+$ and water [51]. Similarly, Rivallan et al. proposed that NO$_2$ is transformed into physisorbed HNO$_3$, which mediates the formation of NO$^+$ ions that exchange zeolite protons [44]. This interpretation is in agreement with the absence of these bands in the reduced sample (B). On the whole, these spectra indicate the activity of oxidized ZP sample in oxidizing NO, thanks to the presence of oxygen atoms, which are likely adsorbed on the surface of small iron oxides.

Finally, the third group of bands in the 1950 to 1700 cm$^{-1}$ region is readily assigned to nitrosylic adducts on Fe$^{2+}$/Fe$^{3+}$ extra-framework sites [44–46]. These bands are very weak on both oxidized and reduced ZP samples and will be described in more detail in the following. Since they are typical of highly dispersed extra-framework iron sites, their low intensity indicates that most iron in this sample is present as large iron oxide particles.

### 2.6.2. Fe$^{2+}$ZP and Fe$^{3+}$ZP Samples

The infrared spectra measured upon NO adsorption on Fe$^{2+}$ZP and Fe$^{3+}$ZP are reported in Figure 7 (C,D and E,F, respectively). In both samples, the set of bands assigned to N$_2$O adsorbed on Fe$^{x+}$ ($x$ = 2, 3) sites is weaker with respect to that observed on the oxidized ZP sample, in agreement with their assignment to surface sites present on small iron oxide clusters. As for the bands assigned to NO$^+$ ions, they have different position and intensity depending on the sample and activation treatment (oxidized or reduced, C,E or D,F, respectively). In detail, the oxidized Fe$^{2+}$ZP sample shows a main component centered at 2141 cm$^{-1}$ (shoulder at 2065 cm$^{-1}$), which is similar to that observed on ZP). As observed on the ZP sample, this band almost disappears after the reduction treatment. On the contrary, a very intense band is observed on oxidized Fe$^{3+}$ZP at 2155 cm$^{-1}$, which decreases but still shows considerable intensity after reduction (spectra E and F). This suggests that the reactivity of NO with adsorbed oxygen strongly depends on the nature of the inserted iron species. Namely, this reaction takes place mostly in Fe$^{3+}$ZP, while oxidized ZP and Fe$^{2+}$ZP have a similar lower activity. The different positions of the NO$^+$ band suggest different local environments, which could be due to vicinal iron and Brønsted sites. This is in agreement with our previous results, showing the presence of Fe(OH)Si Brønsted sites in Fe$^{3+}$ZP [21].

Coming to the nitrosyl region, the oxidized Fe$^{2+}$ZP sample shows a relatively intense peak at 1878 cm$^{-1}$, with shoulders at 1902 cm$^{-1}$ and 1817 cm$^{-1}$ and a very weak band at 1755 cm$^{-1}$ (curves C). A much higher intensity is observed after reduction (curves D). A similar trend with a lower intensity is observed on sample Fe$^{3+}$ZP (curves E). Bands in this spectral region have often been observed on synthetic Fe-zeolites, and are mainly ascribed to highly dispersed Fe$^{2+}$ and Fe$^{3+}$ sites [19,44,45,52,53].

### 3. Discussion

Several types of iron active sites, such as isolated iron cations, binuclear oxygen-bridged Fe species, iron oxide clusters, hydrocyanic acid, isocyanate species, etc., have been reported for the selective catalytic reduction of NO$_x$ [23,54–56], but their precise assignment remains a controversial issue. We consider that such iron active species are heavily dependent on the applied method to incorporate iron, the used iron source, and the homogeneity of the resulting sample. In this report, the most active sample is the Fe$^{2+}$ZP sample, which also shows the highest dispersity of inserted Fe ions, as probed by NO (intensity of the nitrosyl bands in the 1950 to 1700 cm$^{-1}$).

The mechanism and the nature of intermediate species in producing N$_2$ in the selective catalytic reduction of NO$_x$ is also a topic of controversy. Some intermediate species, such as NO$_2$, nitrosyl, nitro, nitrate, NO-NO triplet species, and dissociatively chemisorbed NO, have been proposed. Several studies have outlined that NO$_2$ in particular plays an important role in the reduction of NO$_x$ to N$_2$. It has been reported that NO is oxidized to NO$_2$ with oxygen and then the resulting NO$_2$ reacts with

hydrocarbons to form an active intermediate [53,57–59]. In our case, NO adsorption shows nitrosyl complexes as primary compounds formed in the most active $Fe^{2+}ZP$, $NO^+$ in $Fe^{3+}ZP$, and $N_2O$ in the least active ZP. We can thus tentatively draw a correlation between the abundance of dispersed sites able to add NO as ligand(s) without its disproportionation at RT (bands in the 1950–1700 $cm^{-1}$ region), and the catalytic activity in NO conversion. On the contrary, neither $NO^+$ nor $N_2O$ formation at RT can be directly related to the samples' catalytic activity. In both cases, these features can be associated with the presence of agglomerated particles, in line with early studies by UV-Vis diffuse reflectance and Mössbauer spectroscopies [21].

On the other hand, it is observed that the maximum reduction of NO on $Fe^{2+}ZP$ takes place at 410 °C. After this temperature, NO reduction starts to decrease. According to [53], such a decrement can be associated to a shift in the NO adsorption equilibrium (adsorption is usually an exothermic process) and to the properties of iron species active in NO reduction at higher temperatures.

Finally, the reported results also allow for some semi-quantitative considerations about the extinction coefficients of the bands related to different adsorbed species formed upon NO adsorption. As mentioned above, the sample with higher activity ($Fe^{2+}ZP$) is not only characterized by the highest nitrosyl band intensity, but also the highest NO sorption capacity. This should be, at first approximation, related to the overall integrated area of the adsorbed specie. However, by comparing the infrared results of Figure 7, the highest integrated area among the three samples is observed in $Fe^{3+}ZP$, particularly in the nitrosonium region. Since this does not correspond in a higher NO sorption capacity, we can conclude that $NO^+$ species are characterized by a higher extinction coefficient with respect to nitrosyl bands.

## 4. Materials and Methods

A purified zeolite material was obtained from the zeolitic rock of Palmarito de Cauto deposit, via magnetic and gravimetric purifications [60]. This zeolite is a mixture of ~70% of mordenite with other phases (clinoptilolite-heulandite, montmorillonite, quartz, feldspar, and iron oxides). The chemical composition of the sample in oxide form, with the balance as water, is 61.93% $SiO_2$; 13.42% $Al_2O_3$; 4.95% CaO; 2.53% $Fe_2O_3$; 1.14% MgO; 1.52% $K_2O$; 2.33% $Na_2O$; and 1.22% FeO. This purified zeolite is referred to as the natural mordenite (ZP).

In order to obtain natural mordenite forms modified with iron, ZP samples with a particle size class of +38–74 μm, were exchanged at pH = 2 with $FeSO_4$ and $Fe(NO_3)_3$ 0.05 N solutions. The exchanges were carried out with a solid/liquid ratio of 1 g/20 mL at 80 °C while stirring for 24 h and replacing exchange solutions after 12 h. The resulting samples were washed (firstly with acidified distilled water and then with distilled water) until nitrate and sulfate anions were totally removed and pH was neutral. They were then oven dried at 100 °C and stored in a desiccator. The ZP forms modified with $Fe^{2+}$ and $Fe^{3+}$ were designated as $Fe^{2+}ZP$ and $Fe^{3+}ZP$, respectively.

Elemental analysis of the samples was obtained by flame photometry (Na and K) and atomic emission spectrometry with inductively coupled plasma (Si, Al, Fe, Ca, and Mg) using a Corning 400 photometer and an EPECTROFLAME Modula F spectrometer, respectively. To this aim, the samples were firstly dissolved using a mixture of fluoridric and perchloric acids. Powder X-ray diffraction (XRD) patterns of ZP, $Fe^{2+}ZP$, and $Fe^{3+}ZP$ samples were measured with a D8 Advance Bruker X-ray diffractometer (Bruker AXS GmbH, Berlin, Germany) with Cu-K$\alpha$ monochromatic radiation (1.5406 Å) with 40 kV and a current of 20 mA. The XRD patterns were registered in the 2θ range from 10 to 80° at room temperature (RT). $N_2$ gas adsorption/desorption isotherms were measured by means of a Micromeritics TriStar II device (Norcross, GA, USA) at liquid nitrogen temperature (−196 °C, LNT). Before the sorption measurements, the samples were pretreated in vacuum for 8 h at 350 °C. High resolution transition electron microscope (HRTEM) images of the samples were registered with a JEOL 3010-UHR microscope (Tokyo, Japan) with an acceleration potential of 100 kV, by dispersing the powdered samples on a copper grid covered with a lacey carbon film.

Infrared spectra were recorded on a BRUKER FTIR-66 spectrometer with a resolution of 2 cm$^{-1}$, using an MCT detector. Measurements were carried out using a home-made cell, allowing in situ thermal treatment, gas dosage, and measurement at RT. Thin self-supporting pellets for transmission measurements (around 10 mg/cm$^2$) were prepared with a hydraulic press. Oxidation treatments were carried out by dosing 60 Torr of O$_2$ at 550 °C for 1 h after a heating ramp in vacuum. Similarly, reduction activation was carried out at 400 °C for 1 h with 60 Torr of H$_2$. In both cases, the samples were evacuated before cooling down to RT. NO, carefully distilled, was dosed on the samples at RT after measuring the zeolite-activated reference spectra. Spectra were measured following a stepwise NO pressure (P$_{NO}$) reduction.

Catalytic activity tests of the samples in the selective catalytic reduction of NO with CO/C$_3$H$_6$ were done in a quartz flow reactor, using 100 mg of catalysts, within a 25 to 500 °C temperature interval with a ramp rate of 5 °C/min. The NO (0.09%), C$_3$H$_6$ (0.22%), O$_2$ (0.46%), and CO (1.179%) reaction mixture was prepared by mixing individual flows with mass-flow controllers. A total flow of 55 mL/min was obtained using N$_2$ as a diluent. Effluent gases were analyzed with a CAI ZRE gas analyzer. Before the catalytic test, the samples were pretreated in oxygen flow (0.5% in N$_2$) up to 350 °C with ramp rate of 5 °C/min. The temperature was then decreased to 25 °C and the flow was switched to the reaction mixture.

## 5. Conclusions

Iron-mordenite forms (Fe$^{2+}$ZP and Fe$^{3+}$ZP) were obtained from natural purified mordenite from Palmarito de Cauto (ZP) deposit, Cuba, by hydrothermal ion-exchange processes in acid medium with Fe$^{2+}$ and Fe$^{3+}$ cations. Only small variations were observed in the samples' crystallinity and textural properties after ion exchange, the latter ascribed to the dissolution in the acidic environment of large iron oxide particles from the zeolite external surface with parallel formation of small Fe$_x$O$_y$ clusters within the zeolite pores. In Fe$^{3+}$ZP, a shift in the diffraction peaks' angular position was observed, which corresponds to an increment in the inter-planar distance and expansion of the zeolite matrix due to the incorporation of Fe$^{3+}$ into the mordenite framework.

The activity in HC-SCR (used as a reaction test) was found to be dependent upon the ion exchange process, with Fe$^{2+}$ZP showing the highest catalytic activity, followed by Fe$^{3+}$ZP and ZP samples. The same order Fe$^{2+}$ZP >> Fe$^{3+}$ZP >> ZP was also observed in the NO sorption capacity of the samples at low temperatures (<170–200 °C).

In agreement with these results, FTIR-NO studies showed a higher intensity of the nitrosyl bands formed upon NO adsorption on the Fe$^{2+}$ZP sample. These bands (present on both oxidized and reduced samples) are related to isolated/highly dispersed Fe$^{2+}$/Fe$^{3+}$ ions in extra-framework positions, which are more abundant in Fe$^{2+}$ZP and are responsible for the high sorption capacity (and likely activity) of this sample. On the contrary, the intensity of nitrosonium ions NO$^+$, formed upon reaction of NO with oxygen and likely protons, and more abundant on Fe$^{3+}$ZP sample, could not be related to neither NO sorption capacity or activity.

**Author Contributions:** Conceptualization, F.C.R. and G.B.; investigation F.C.R., D.T.F. and B.C.-R.; resources V.P.; data curation, F.C.R., I.R.-I. and G.B.; writing—original draft preparation, F.C.R., I.R.-I. and G.B.; writing—review and editing, F.C.R., I.R.-I. and G.B.

**Funding:** This research was partially supported by UNAM-PAPIIT through grant IN107817 and grant Russia-Cuba 18-53-34004.

**Acknowledgments:** F.C.R. acknowledges support from COFAA-IPN-México as well as technical support from Marco Fabbiani of the Department of Chemistry in Torino, Italy. Thanks are given to E. Aparicio, I. Gradilla, J. Peralta, J. Gonzalez and E. Flores for technical assistance.

**Conflicts of Interest:** The authors declare no conflict of interest.

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
