# Peer review of "Fe Speciation in Iron Modified Natural Zeolites as Sustainable Environmental Catalysts"

_catalysts, doi:10.3390/catal9100866_

Round 1
Reviewer 1 Report
This paper reported catalytic activities of natural mordenite and Fe ion exchanged ones toward NO reduction. They characterized physical-chemical properties of the zeolites by using BET, XRD, TEM, and IR, and so on. I think present data were meaningful as references for the readers in associated field. However, the manuscript was a bit wordy, and many typos were found throughout manuscript. Below is my detail comments on this draft.
1. Please check grammars and typos in the manuscript. There were many errors.
*pp2 line 43, avoding – avoiding line 83, whit – with, etc…
2. In page 3, line 120-127
Could you add comments why natural mordenite exhibited lower CO and C3H6 oxidation, compared with Fe3+ ZP? If the oxidation came from simple oxidation by O2, there should be comparable generation of CO2 in case of natural mordenite.
3. In page 4, line 134-148
I can see some diffraction peaks, not originated from mordenite in Figure 3. Can you identify what they are? Since it is unique natural zeolite, but not pure mordenite, I think it needs more precise description of crystal structure including information of crystalline impurities.
4. In page 5, line 155-159
Based on BET as well as TEM images, ion exchange steps might dissolved mordenite crystals generating mesopores, which eventually increased surface area but decreased micropore volume. How do you know ion-exchange step only removed impurities phase? And, what are the soluble impurities? Please be specific.
5. In page 7, pp 192-205
Considering the unit cell size of mordenite, crystal sizes of 2.5 nm and 5 nm shown in Figure 6 were not believed to be mordenites, which never reported in the literature? Show more evidence. Other metallic crystals could have a lattice distance around 0.41nm.
6. In page 6, FTIR data(Figure 7, 8, 9).
I think it is better idea to put Figure 7, 8, 9 together into a figure for easy comparison. Please rewrite the part with more concise manner. It was too wordy.
Reviewer 2 Report
This manuscript by Rodríguez-Iznaga, Berlier et al. describes the preparation of Fe-ion doped natural zeolites via a thermal treatment and the catalytic processes for NO conversion by the resulting materials. The results indicate that the Fe(II)-doped ZP has the highest affinity to NO adsorption while the Fe(III)-doped ZP exhibits the best catalytic performance for the conversion of NO to CO2. The FTIR measurements are conducted on the gas molecules attached zeolite materials, showing that the major species are the Fe-N2O, Fe-NO and Fe-NO+ in ZP, Fe(II)-ZP and Fe(III)-ZP, respectively. The assignments are made according to previously report results. It is hard to find the correlation between the reactivity and selectivity of any specific materials and the FexOy aggregates. There are a few questions to be answered prior to acceptance of the manuscript.
(1) There are no data presented in the manuscript to support the oxidation state of the doped Fe sites in Fe(II)-ZP and Fe(III)-ZP. Can the author provide spectral evidences such as X-ray absorption spectroscopy, XPS, etc. to warrant the assignment of the Fe oxidation state?
(2) It is described that the attached NO on the Fe(III)-ZP reacts with O2 and protons to the NO+ and H2O via an intermediate species, NO2. The interconversion of NO-NO2-N2O-NO3- is an interesting topic and has been under extensive investigation for a long time. Oxygenation of NO to NO2 can be occurred by the reaction of oxygen or the self-combination of NO molecules. Was the reaction conducted under N2 available?
(3) In order to elucidate the mechanism, isotope experiments are recommended.
(4) What is the conversion efficiency for each system?
